cognition, neuroscience

navigation, backtracking, dACC, alpha

**Authors for correspondence:**
Eva Zita Patai
e-mail: e.patai@ucl.ac.uk
Hugo J. Spiers
e-mail: h.spiers@ucl.ac.uk

†These authors contributed equally to this study.

# Backtracking during navigation is correlated with enhanced anterior cingulate activity and suppression of alpha oscillations and the 'default-mode' network

Amir-Homayoun Javadi[1,3,†], Eva Zita Patai[1,†], Eugenia Marin-Garcia[1,4], Aaron Margois[2], Heng-Ru M. Tan[1], Dharshan Kumaran[2,5], Marko Nardini[6], Will Penny[7], Emrah Duzel[8], Peter Dayan[9] and Hugo J. Spiers[1]

[1]Institute of Behavioural Neuroscience, and [2]Institute of Cognitive Neuroscience, University College London, London, UK
[3]School of Psychology, University of Kent, Canterbury, UK
[4]School of Psychology, University of the Basque Country (UPV/EHU), Bilbao, Spain
[5]Google Deepmind, London, UK
[6]Department of Psychology, Durham University, Durham, UK
[7]School of Psychology, University of East Anglia, Norwich, UK
[8]Institute of Cognitive Neurology and Dementia Research (IkND), University Hospital Magdeburg, Magdeburg, Germany
[9]Max Planck Institute for Biological Cybernetics, Tübingen, Germany

EZP, 0000-0003-2589-2239

Successful navigation can require realizing the current path choice was a mistake and the best strategy is to retreat along the recent path: 'backtrack'. Despite the wealth of studies on the neural correlates of navigation little is known about backtracking. To explore the neural underpinnings of backtracking we tested humans during functional magnetic resonance imaging on their ability to navigate to a set of goal locations in a virtual desert island riven by lava which constrained the paths that could be taken. We found that on a subset of trials, participants spontaneously chose to backtrack and that the majority of these choices were optimal. During backtracking, activity increased in frontal regions and the dorsal anterior cingulate cortex, while activity was suppressed in regions associated with the core default-mode network. Using the same task, magnetoencephalography and a separate group of participants, we found that power in the alpha band was significantly decreased immediately prior to such backtracking events. These results highlight the importance for navigation of brain networks previously identified in processing internally-generated errors and that such error-detection responses may involve shifting the brain from default-mode states to aid successful spatial orientation.

## 1. Introduction

When navigating familiar environments animals can call upon spatial memories to flexibly adapt routes between locations of interest [1]. For example, we can take a novel route to a restaurant or we may be forced to take an alternative route owing to some roadworks. Such flexible behaviour during navigation is thought to rely on the 'cognitive map' [2]—an internal representation of the environment. However, sometimes after beginning a journey it is possible to realize that a poor route choice was made or that the route chosen is not going to reach the goal. In these instances, a good strategy is to turn around and backtrack to a previous point and begin an efficient route. Such spontaneous and

internally-generated behaviour is anecdotally common, and reports of human behaviour that include backtracking often discuss it in the context of lost individuals [3], specifically how backtracking is a part of explorative behaviour and the flexible use of landmarks [4] ('looking back' has been shown to aid subsequent navigation by allowing the navigator to see the environment from different perspectives, i.e. forming an allocentric map), and how the choice to initiate a backtrack is related to confidence levels [3]. Additionally, backtracking is a strategy that is not only common but integral to navigation in visually impaired people [5], and even sighted people have been reported to dislike when not given the option to backtrack [6]. It is not clear if backtracking behaviour is related to navigation ability [7,8], though older people do show a deficit in retracing their steps [9], and navigators often return to a previous decision point and perform a breadth-first search of spaces—i.e. of upcoming the path options [10], especially if they are skilled [3]. Finally, this behaviour may be universal in animal navigation, as it has also been reported in ants, which use backtracking as part of specific search strategy when displaced from their nest [11], and is implemented in many robotics applications [12]. Despite the wealth of observational reports on backtracking, how the brain supports such backtracking behaviour during goal-directed navigation remains elusive, and it is unclear under what conditions this (usually) corrective behaviour arises.

To explore the human brain regions engaged in backtracking as well as the fine-grained temporal neuronal dynamics, we combined functional magnetic resonance imaging (fMRI), as well as magnetoencephalography (MEG), with a virtual reality (VR)-based environment ('LavaWorld') in which participants navigated a desert island containing hidden treasure (goal objects) with paths constrained by lava, which had the capacity to shift and open new paths (shortcuts) or close others (detours). In prior analysis of this data we examined the neural responses to changes in the maze layout—detours and shortcuts [13]. Here we examine spontaneous backtracking events with the prediction that brain regions associated with planning, self-driven error-correction and re-engagement in the ongoing task will be involved. Under the hypothesis that re-orienting during backtracking would involve re-engagement in the task we predicted a reduction in default-mode activity [14]. Based on prior studies reporting self-correction of errors ([15,16]; for review see [17]) we predicted prefrontal and anterior cingulate regions would show increased activity during backtracking. Relatedly, one study found that the right posterior hippocampus was engaged when a switch in the path was needed at forced detours [18], thus it is possible that backtracking may engage this region owing to re-estimation of the future path (see [19]).

## 2. Methods

### (a) Participants

fMRI. Twenty-two subjects (mean age: 21.8 ± 2.3 years, range: 19–27; 14 female). To avoid testing participants with poor navigation skills, participants were administered a questionnaire regarding their navigation abilities/strategies (Santa Barbara Sense of Direction Scale; mean score = 4.9, range: 3.7–5.7). MEG. Twenty-five subjects (mean age: 22.5 ± 3.9 years, range: 18–31; 12 female). Participants were administered a questionnaire regarding their navigation abilities/strategies (Santa Barbara Sense of Direction Scale; mean score = 5.1, range: 3.2–6.8).

There was no overlap in participants between the fMRI and MEG tasks. All participants had normal to corrected vision, reported no medical implant containing metal, no history of neurological or psychiatric condition, colour blindness, and did not suffer from claustrophobia. All participants gave written consent to participate in the study in accordance with the Birkbeck-UCL Centre for Neuroimaging ethics committee. Subjects were compensated with a minimum of £70 plus an additional £10 reward for good performance during the scan. One participant was excluded from the final sample because there was severe signal loss from the medial-temporal area in their functional scan.

### (b) Virtual reality environment: LavaWorld

A virtual island maze environment was created using Vizard virtual reality software (© WorldViz). The maze was a grid network, consisting of 'sand' areas that were walkable, and 'lava' areas, which were unpassable and as such were like walls in a traditional maze. However, the whole maze layout was flat, so there was visibility into the distance over both sand and lava. This allowed participants to stay oriented in the maze throughout the task. Orientation cues were provided by four unique large objects in the distance. Movement was controlled by four buttons: left, right, forwards and backwards. Pressing left, right or backwards moved the participant to the grid square to the left, right or behind respectively (if there was no lava in the way), and rotated the view accordingly. Similarly, pressing forward moved the participant to the next square along. See figure 1 for a participant viewpoint at one point in the maze. Participants were tested over 2 days, on day one they were trained on the maze, and on day two they were tested on the maze in the MRI/MEG scanner.

### (i) Training

On the first day, participants were trained on the virtual maze (25 × 15 grid) to find goal locations. During this phase, all goal objects (20 in total, distributed across the maze) were visible at all times, and participants navigated from one to the next based on the currently displayed target object (displayed in the top-right corner of the screen). After 1 h of training, subjects were given a test to establish how well they had learnt the object locations. On a blank grid, where only the lava was marked, participants had to place all the objects they remembered. They were given feedback from the experimenter, and if needed, prompts as to the missing objects. This memory-test was repeated twice more during the training, after 1.5 and 2 h. At completion, for participants to return for the fMRI/MEG phase on the second day, they had to score at 100% accuracy in placing the objects.

### (ii) Navigation test and functional magnetic resonance imaging/ magnetoencephalography scan

On the test day, participants were given a brief refresher of the maze with the objects before beginning the test phase. Before scanning, participants were allowed to familiarize themselves with the scanner button pad, and the changes that would occur. This involved presenting them with a novel environment that had not been experienced on day one, and which had no objects, different distal cues and a different maze layout, to avoid any confounds or confusion with training and test mazes. Participants could then practise the task in this new environment, and accustom themselves to the controls (button pad with four active buttons: left, right, forward, and turn around) and to the appearance of changes to the lava.

While in the MRI/MEG scanner, participants performed the test phase of the experiment. A single trial in the test phase is

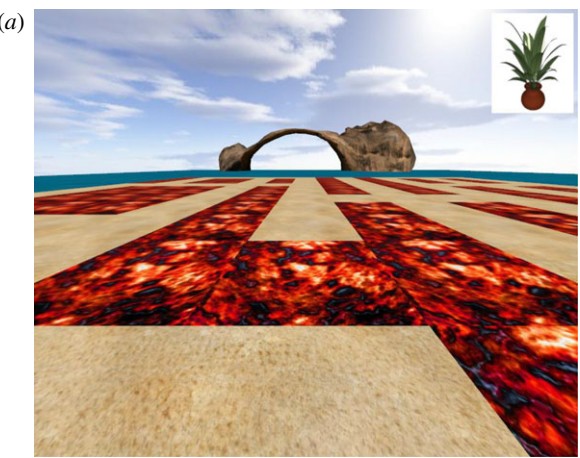

(a)

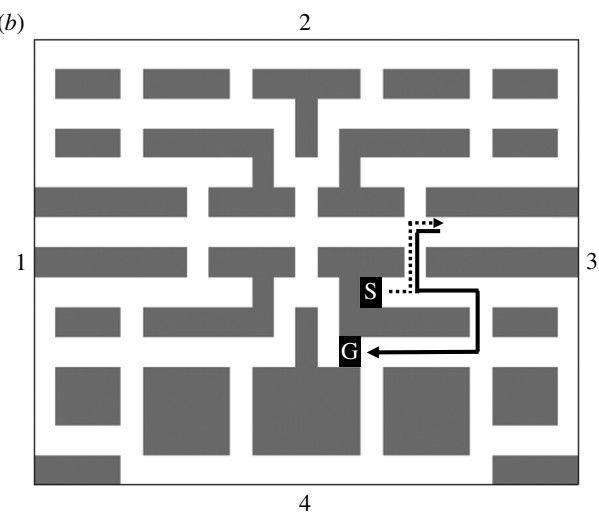

(b)

- **S** start location
- **G** goal location
- ▪▪▪▶ backtracking point
- ▪▪▪ path prior to backtrack
- ▬ path after backtracking
- ▪ lava
- ▫ walkable paths

**Figure 1.** LavaWorld. (a). Example view of test environment during scanning and current goal object (top right corner). A distal cue is visible (arch), and three others were located at the other cardinal directions. The sand represents the path that can be moved along, whereas the red 'lava' blocks in the path. During training, objects (20 in total) were visible across the whole maze, and participants used the controls to move forward, left/right and backward to collect them, with an arrow guiding them towards the object (in the first of three rounds of training). During the test phase, the 20 objects were *not* visible and the environment could change, such that the lava shifted around to close an existing path, or reveal new paths. (b) Bird's eye view of the maze from above, with the four cardinal landmarks marked by numbers, and a hypothetical example of a backtracking event. This example route is one of 30 (on average) routes that would be experienced during scanning. (Online version in colour.)

defined as being informed of the identity of the new goal object, then finding the way to, and arriving at, it. During the test phase, two things were different from training: (i) target objects were not visible, so participants had to navigate between them based on their memories of the locations, and (ii) the lava could move, blocking some paths and creating new paths. Participants were informed that this was a temporary change and that after reaching the goal the environment would revert to the baseline state. During each journey to an object, a single change event occurred in the lava layout. At the point of a change, the screen froze for 4 s to ensure that participants had an opportunity to detect the change and consider their path options. These changes could either be detours (when a piece

of lava was added to block the current path on the grid, thus forcing the participant to take an new longer route to their goal); shortcuts (a piece of lava was removed and replaced with sand, allowing the participant to pick a shorter route); false shortcuts (visually identical to shortcuts, but choosing a route through them would increase the net distance to the goal because of the layout of the maze; false shortcuts came in two classes: false shortcuts towards and false shortcuts away from the goal, depending on whether or not the false shortcut seemed to lead in the general direction of the goal or if it was an opening pointing away from it, see [13] for visual examples); and a condition in which the screen froze, but no lava was added/removed. For detours and shortcuts, there were also two levels of change to the (optimal) new path, either four or eight grid steps extra/less, respectively. Finally, during the test phase there were also control 'follow' trials which started with an arrow that indicated the direction to travel. In this case, participants were required to follow the twists and turns of the arrow until a new target object appeared, and from then onwards the trial was like the 'navigation' events as described above.

Participants performed a total of 120 routes, with one change event occurring in each route (number of trials per condition was 17 on average, range: 11–25, depending on the different scenarios used for counterbalancing routes taken). Each route started from a previous goal and ended at the new goal object for that trial.

For this analysis, we were specifically interested in spontaneous backtracking events, which were defined by when participants pressed the backwards button to turn and return to a step along the route they had just come down. Non-backtracking events were picked from the participants' other paths and were matched to these in the relative number of steps taken before the backtracking happened (for example: halfway through the route). We did not use the absolute number of steps, because trials that contained a backtracking event were often much longer, and thus the step number at which a backtracking event occurred could have already been at (or past) a goal point in a non-backtracking trial. We used step matching rather than elapsed time matching because participants' speed was not controlled and we wanted to match the events based on actual navigation steps and not potential time differences owing to stationary behaviour. This resulted in around 30 events per participant ($30.5 \pm 2.9$ s.e.m.). For follow-up analyses we also looked at onset of turn (and non-turn) events, which for one analysis were taken as all such events in the experiment (and subsampled to match one another in trial numbers), and another model in which we took these events from the non-backtracking trials and thus matched event numbers with this condition. As a follow-up, we also took turns from backtracking trials only, (in order to have comparative distribution among different trial types). Finally, we also took a new set of non-backtracking events from the same trials as those in which backtracking did occur, excluding any steps near the beginning, end, or near the change point. These are helpful in providing evidence that the effects observed for backtracking were, for example, not simply driven by comparing events sampled in periods after forced-detours with other events sampled in trials with no detours.

## (c) Functional magnetic resonance imaging scanning and analysis

Scanning was conducted at the Birkbeck-UCL Centre for Neuroimaging (BUCNI) using a 1.5 Tesla Siemens Avanto MRI scanner (Siemens Medical System, Erlangen, Germany) with a 32 channel head coil. Each experimental session lasted around 60 min and was separated in three parts (each of approximately 15–20 min). The details of the functional scan can be found in Javadi *et al.* [13]. A T1-weighted high-resolution structural scan was acquired after the functional scans (repetition time = 12 ms, echo time = 5.6 ms, $1 \times 1 \times 1$ mm resolution). Ear plugs were used for noise

reduction, foam padding was used to secure the head in the scanner and minimize head movements. Stimuli were projected to the back screen, a mirror was attached to the head coil and adjusted for the subjects to see full screen. All fMRI preprocessing and analysis was performed using SPM12. To achieve T1 equilibrium, the first six dummy volumes were discarded. During preprocessing, we used the new Segment (with six tissue classes) to optimize normalization. Otherwise, we used all default settings, and we performed slice timing correction. No participants had any abrupt motion change over 4 mm.

General linear models were constructed at the onset on the backtracking (and non-backtracking, turn and non-turn) events, with a duration of 0 s. We report results surviving family-wise error correction (FWE), as well as results of region-of-interest (ROI) analysis of the putative default-mode network (DMN) (defined anatomically including the medial prefrontal cortex (PFC), the precuneus/posterior cingulate cortex, bilateral parahippocampal cortex and angular gyrus). We obtained the ROI mask from Kaplan *et al*. [20] in order to test for involvement of the dorsal anterior cingulate area (dACC). To measure hippocampal response, we used the mask used in our previous study [21].

## (d) Magnetoencephalography recording and analysis

Recordings were made using a 275-channel CTF MEG system with superconducting quantum interference device (SQUID)-based axial gradiometers (VSM Med-Tech) and second-order gradients in a magnetically shielded room. Neuromagnetic signals were digitized continuously at a sampling rate of 480 Hz and then band-pass filtered in the 0.1–120 Hz range. Head positioning coils were attached to nasion, left, and right auricular sites to provide anatomical coregistration to a template brain. Preprocessing and analysis of MEG data was done using FIELDTRIP [22]. Independent-component analysis (ICA) was performed on the continuous data, leading to the identification of blink, saccade and cardiac components, which were removed. MEG data were subsequently parsed into epochs around 'backtracking' events (−1000 to 1000 ms), which were defined as steps on which participants turned around and thus spontaneously changed their paths.

To analyse the MEG data we focused on event-related fields (ERFs), as well as time-frequency analysis. However, owing to the nature of the task (free-viewing during navigation), and despite the ICA correction, we were unable to exclude fully the possibility that some oscillatory signatures would be contaminated by eye-movements. We therefore looked at the difference between the saccade variance as measured by ICA across different conditions, and found that backtracking was not significantly different from non-backtracking.

Given the exploratory nature of this study, we investigated effects of change type using all sensors, and all time points the whole time period. Here we report significant effects found, cluster corrected for multiple comparisons. over frequency ranges based on *a priori* bands as previously reported in the literature (3–7 Hz for theta [23–25], 8–12 Hz for alpha, and 15–25 Hz for beta). Note that for the theta band, we also confirmed this frequency band by measuring peak activity during an orthogonal period (power at the start of the trial comparing goal objects to follow arrows, and found the group peak was at 5.2 (±0.4 s.e.m.) Hz). We did not perform source localization on our MEG dataset as we did not have any structural MRIs for realignment and no detailed headshape model available.

## 3. Results

### (a) Behaviour

Our primary measure of navigation was the accuracy of the whole route, in other words whether participants took the optimal path to the target. We conducted a repeated measures ANOVA to test for effects of terrain change type (detour, shortcut etc.) on participants' accuracy in finding the correct path. We found a significant effect of change type in both fMRI and MEG tasks (fMRI: $F_{1,20} = 35.03$, $p < 0.001$; MEG: $F_{1,23} = 13.04$, $p < 0.001$), a significant effect of magnitude in the fMRI task only (fMRI: $F_{1,20} = 9.77$, $p = 0.005$; MEG: $F_{1,23} = 3.61$, $p = 0.07$), and a significant interaction in both fMRI and MEG (fMRI: $F_{1,20} = 8.15$, $p = 0.01$; MEG: $F_{1,23} = 8.87$, $p = 0.007$), for details see [13].

To follow up the errors in which participants did not take the optimal path, we looked at the number of extra steps taken on a route. We found a significant effect of terrain change type (fMRI: $F_{1,120} = 8.3$, $p < 0.001$, MEG: $F_{1,23} = 35.3$, $p < 0.001$), with overall more steps off-route in the detours (+8) and false shortcuts towards conditions. When quantifying the number of extra steps as a proportion of the total (new) number of optimal steps from the terrain change point onwards for a given route, in fact shortcuts resulted in the largest proportion off-route (table 1). Some of these extra steps were owing to participants turning around, i.e. 'backtracking'; these were again more common in the detours (+8) condition (fMRI: $F_{1,120} = 8.5$, $p < 0.001$, compared to all $t_{1,20} > 2.1$, $p < 0.051$; MEG: more common in the false shortcuts towards the goal and the detours (+8) condition (both compared to all other events $t_{1,23} > 2.1$, $p < 0.046$, but not different from each other $t_{1,23} = 1.0$, $p = 0.32$)). Moreover, we calculated the ratio of correct, compared to incorrect, backtracking trials ('correct' is defined as a trials in which backtracking would actually bring the participant closer to the goal), and found that overall 68% (±8% s.e.m., range: 0–100%; MEG: 77% ± 7.6% s.e.m., range: 0–100%) of backtracking events were correct or optimal, and occurred equally frequently across all conditions ($F_{1,5} = 1.08$, $p = 0.38$; MEG: $F_{1,5} = 0.077$, $p = 0.99$). Thus, in the majority of backtracking events participants became aware that they were heading away from the goal and spontaneously decided to turn around.

### (b) Imaging results

Full results pertaining to external changes, such as detours and shortcuts, are reported in [13].

### (i) Prefrontal responses evoked during spontaneous backtracking events

We were interested in the brain areas that were activated by spontaneous, internally generated route changes, i.e. backtracking events. To investigate their neural correlates, we compared moments in routes where a backtracking event occurred (defined as a return to a previous grid point along a single journey towards a target), to equivalent events where no backtracking happened (matched according to relative steps along a journey). When comparing backtracking to non-backtracking events, we found that during backtracking, activity in a region of medial PFC survived whole-brain FWE correction (figure 2). This activation overlapped with the dACC reported by a recent study when difficult navigational decisions were required [20] as well as when U-turns were required to optimally reach a goal [26]. Follow-up ROI analysis revealed a significant effect in the dACC (small-volume correction: $Z = 5.91$; $p < 0.001$), and no significant effects in the hippocampus. When controlling for trial type (detour, shortcut, etc.), i.e. looking at backtracking

**Table 1.** Behavioural results summary (mean (±s.e.m.)).

| | detour (+8) | detour (+4) | shortcut (−8) | shortcut (−4) | false shortcuts towards | false shortcuts away |
|---|---|---|---|---|---|---|
| **fMRI** | | | | | | |
| correct route choice (%) | 64.1 (±3.9) | 80 (±2.3) | 84.6 (±2.6) | 84.6 (±2.8) | 65.8 (±3.1) | 81.3 (±1.9) |
| extra steps (proportion of path) | 1.2 (±0.11) | 1.4 (±0.6) | 2.1 (±1.07) | 1.7 (±0.37) | 1.3 (±0.18) | 1.4 (±0.32) |
| backtracking (no.) | 7.1 (±1.3) | 3.7 (±0.7) | 2.2 (±0.6) | 2.6 (±0.6) | 4.3 (±0.6) | 3.5 (±0.4) |
| correct backtracking (%) | 71 (±8.6) | 83 (±6.9) | 73 (±9.3) | 65 (±9.8) | 65 (±9.9) | 61 (±10.1) |
| **MEG** | | | | | | |
| correct route choice (%) | 70.2 (±4.5) | 82.2 (±2.3) | 84.8 (±3.4) | 84.2 (±2.8) | 71.1 (±4.2) | 87.1 (±3.1) |
| extra steps (proportion of path) | 1.2 (±0.5) | 1.3 (±0.5) | 2.0 (±1.7) | 1.9 (±1.0) | 1.4 (±0.8) | 1.6 (±1.4) |
| backtracking (no.) | 3.5 (±0.6) | 2.1 (±0.4) | 1.1 (±0.3) | 1.9 (±0.4) | 3.0 (±0.5) | 1.8 (±0.5) |
| correct backtracking (%) | 72.4 (±8.4) | 74.1 (±9.6) | 84.5 (±8.6) | 71.6 (±10.4) | 75.6 (±8.5) | 88.7 (±7.7) |

compared to non-backtracking events within the same trial, we replicate the significant effect of the dACC ROI (small-volume correction: $Z = 4.6$; $p < 0.001$).

### (ii) Backtracking events disengage from the putative default-mode network

Because backtracking involved a visual change in view (180°) compared to non-backtracking events we explored whether similar regions would be active when participants turned left or right compared to when they did not turn. We found a significant response in the pre-supplementary motor area, when comparing turns to non-turns, but no significant clusters in the dACC (see the electronic supplementary material, table S2 for significant clusters). When backtracking was compared to turns the dACC was significantly more active in backtracking events ($Z = 4.45$, $p = 0.007$), in which an equivalent amount of visual change is contrasted (90° difference in change of viewpoint). Conversely, at an uncorrected threshold ($p = 0.001$, min. 5 voxels), we found turns compared to backtracking resulted in responses in the hippocampus, anterior medial PFC and posterior cingulate cortex. These regions are overlapping with those implicated in the DMN [27]. To explore whether the results match the DMN more explicitly we created an aggregate ROI including the medial PFC, the precuneus/posterior cingulate cortex, bilateral parahippocampal cortex and angular gyrus. We found that backtracking significantly suppressed this putative DMN compared with turns ($Z = 4.62$, $p = 0.006$, and $Z = 4.37$, $p = 0.018$ for comparison within backtracking trials only), and this was also the case when comparing backtracking to non-backtracking events ($Z = 4.21$, $p = 0.038$, and using within trial control: $Z = 4.77$, $p = 0.003$), but not when comparing non-turns to turns ($p > 0.1$), highlighting disengagement from this network when participants spontaneously instigated a route change (figure 3; electronic supplementary material, table S2). When comparing turns and non-turns, with only matched number of events (to backtracking), we find no significant effects between the conditions, further underscoring the magnitude of the backtracking effects seen despite limited trial numbers.

### (iii) Oscillatory dynamics during backtracking

We considered that backtracking would probably require a change in the allocation of attentional resources in order to select a new path. Such an allocation of attention to select behaviourally relevant items among competitors, both externally [28] or from memory [29], has consistently been linked to a reduction in alpha band (8–12 Hz) power. Thus, we hypothesized that alpha power might reduce during backtracking events compared to control events. Consistent with our prediction we indeed found a significant decrease in alpha power preceding the onset of backtracking compared to non-backtracking events (−700 to −150 ms, $p = 0.008$), over a wide range of posterior sensors (figure 4). Control frequency bands (theta, 3–7 Hz; and beta, 15–25 Hz) did not reveal any effects in the pre-backtracking time-window. Moreover, the change in alpha activity could not be explained by differences in saccadic activity across the trial types (whole trial or during the significant pre-backtrack time-window, $t_{1,23} < −0.9$, $p > 0.1$). There was no correlation between the trial-wise alpha decrease and subsequent reaction times (i.e. time between button presses, as this was spontaneous

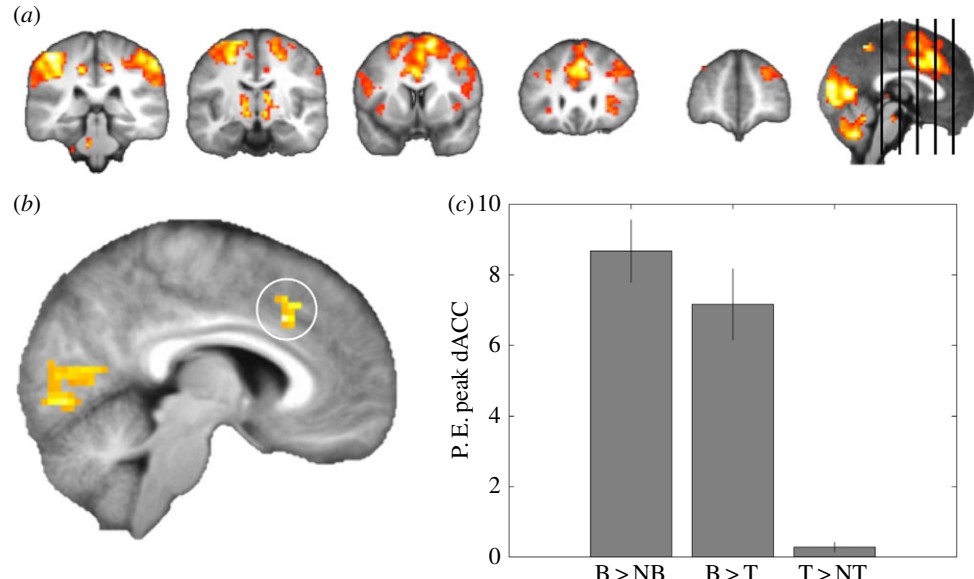

**Figure 2.** Backtracking activates a range of frontal areas, as well as the dorsal anterior cingulate cortex (dACC). (*a*) The superior frontal gyrus and right lateral PFC are activated during backtracking compared to non-backtracking events. Figures are thresholded at *p* = 0.005 uncorrected. (*b*) Whole-brain results (FWE *p* < 0.05) revealed a significant activation in the dorsal anterior cingulate cortex. (*c*) Extracting parameter estimates (P.E.) from the peak voxel in dACC (MNI: *x*: 6, *y*: 20, *z*: 35, note Kaplan *et al*. [20,23] = *x*: 6, *y*: 23, *z*: 37) show that backtracking activates this region significantly. Bar plot is for illustration purposes only to highlight the effect, error bars represent s.e.m. B, backtracking; NB, non-backtracking; T, turns. (Online version in colour.)

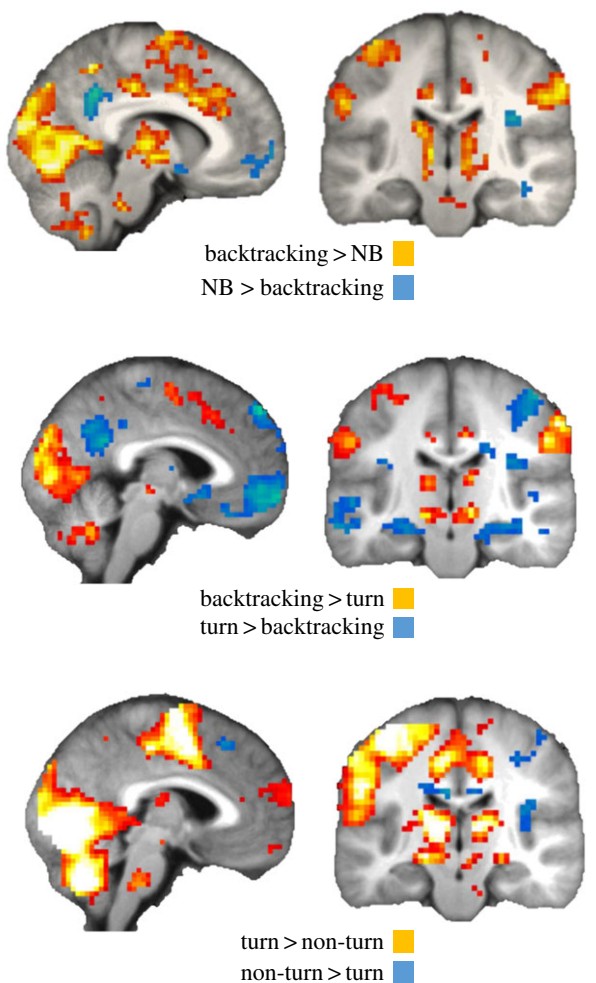

**Figure 3.** Backtracking events disengage regions in the putative default-mode network (DMN) when compared to other events. Results are shown at *p* = 0.005 uncorrected, minimum five contiguous voxels. Significant results in the combined DMN mask: backtracking compared to non-backtracking (*p* = 0.038, *Z* = 4.21), and turns (*p* = 0.006, *Z* = 4.62). NB, non-backtracking. (Online version in colour.)

behaviour) to choose to backtrack ($r < 0.03$, $p > 0.1$). To confirm the robustness of our results, we also investigated the decreased alpha power in backtracking in only those participants who had a minimum of five such events ($n = 21$), and found the effect replicated in a similar time-window ($-600$ to $-10$ ms, $p = 0.034$). We also checked whether there was a modulation in alpha power before participants made any type of turn (i.e. left or right, but no backtracking), and found no significant effect, indicating that the process of backtracking selectively involves allocation of attentional resources and is not a confound of visual activity induced by direction changes. Additionally, when directly comparing backtracking to simple turning events, there was a significant decrease in alpha power in the pre-step time-window ($p = 0.005$), further underscoring the notion that there are differences in attentional resource allocation during backtracking specifically.

## 4. Discussion

In this study we report the neural correlates of spontaneous 'backtracking' behaviour. We found increased activation in the dorsomedial PFC, an area that has been implicated in various contexts including: navigational planning [30,31], hierarchical planning [26], high planning-demand decisions [20], and model updating, irrespective of difficulty or simple error-related signalling [32–34]. More specifically, medial PFC correlated with a model-based estimate of behaviour relating to putative events of re-tracing a path in a simple maze where the task was to self-localize by exploring [35]. Our peak activation for the dACC is highly proximal to the one previously reported by Yoshida & Ishii [35]. This highlights the role of the medial PFC/dACC in situations in which re-evaluation and updating from internal monitoring are required, as is the case during introspective awareness [36], and reorienting and error detection [37,38]. The

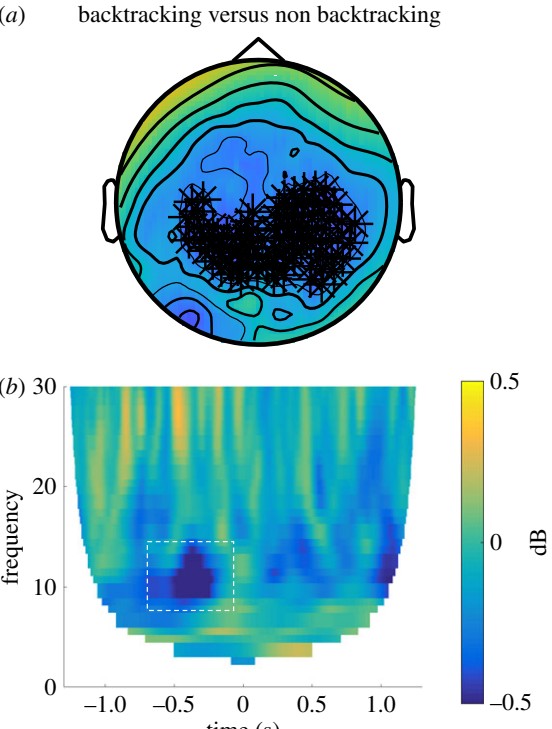

(a) backtracking versus non backtracking

(b)

**Figure 4.** Alpha power decreases prior to backtracking events. Preceding spontaneous path changes, i.e. backtracking, we found decreased alpha power over posterior sensors (−700 to −150 ms before the backtracking event). This effect was specific to the alpha band (no significant effects for theta or beta band in the same period). Top panels show topographies of significant sensors. Bottom displays time-frequency plots of the data from all significant sensors, showing the specificity of frequency bands as well as temporal profile of condition differences (significant frequency/time-period highlighted with dashed box). (Online version in colour.)

activation of the dACC possibly reflects stronger engagement of planning and error signals when participants spontaneously realized that they were on a sub-optimal path, which is further supported by the result that nearly 70% of backtracking events were usefully corrective, resulting in a shorter path to the goal. In addition to the dACC, the saliency network—independent of task-specifics (cognitive, emotional etc.)—has been described as including the orbitofrontal insula [39,40], which we also found during backtracking (electronic supplementary material, table S2), underscoring the notion that a change in personal salience or introspective awareness [36,41] occurred during these events.

Additionally, when comparing backtracking to non-backtracking, or turn events, we found a disengagement from the putative DMN (including the medial PFC, bilateral hippocampi, posterior cingulate cortex), consistent with the notion that internally generated route changes resulted in a global resetting of attention from internal to external sources, which may be specifically orchestrated by the dACC and insula [40]. The DMN has been implicated during transitions between tasks/states and at task restarts [14,42,43], when participants were 'in the zone' during a self-generated behaviour [44], and before an error is committed [45].

Finally, immediately preceding backtracking, we observed a decrease in alpha power, consistent with changes in the allocation of attention (for review see [46]), which has been argued to be important during navigation [47,48]. Notably, a previous study also found decreased alpha power when changes occurred to direction of passive virtual movement in navigation [49]. We show that this suppression is also present for changes initiated by the person actively navigating and that these putative markers of attentional selection precede the onset of the direction change. This result may be important in light of previous reports showing a positive relationship between power in the alpha frequency band and DMN activity [50,51] as we found here: backtracking disengages from the DMN and reductions in alpha power occur prior to backtracking. This result highlights the important role of attention shifts during navigation, from putatively internal to external sources, as well as the usefulness of using different methodologies to understand the neural mechanisms of behaviour.

Here we designed our task to explore flexible navigation in a maze with a dynamic terrain, and discovered patterns of spontaneous backtracking in the trajectories. As such, we were able to compare backtracking against both non-backtracking events and turns made in the maze. While this allowed us to characterize the response to backtracking (e.g. turns do not evoke prefrontal responses or a change in alpha power observed in backtracking) it is important to acknowledge that future research will be needed to determine whether the neural responses in different regions during backtracking are driven by updating route-plans, setting sub-goals, suppressing prior actions, visual information, and/or the novelty of the responses.

## 5. Conclusion

Flexible planning and reorienting is essential for successful navigation. Here we report on backtracking, events that were spontaneous and were not planned for experimentally. Exploring spontaneous human behaviour during naturalistic tasks such as navigation, including error monitoring and updating, could lead to more applications for cross-species comparisons of general problem-solving behaviour.

**Ethics.** All participants gave written consent to participate in the study in accordance with the Birkbeck-UCL Centre for Neuroimaging Ethics Committee. Subjects were compensated with a minimum of £70 plus an additional £10 reward for good performance during the scan.

**Data accessibility.** The datasets supporting this article are available from the Dryad Digital Repository: https://doi.org/10.5061/dryad.v49j1gs [52].

**Authors' contributions.** A.-H.J designed and conducted study. E.Z.P. analysed the data and wrote the manuscript. E.M.-G. was involved in data analysis. A.M. and H.-R.M.T. were involved in data collection. D.K., M.N., W.P. and E.D. were involved in study design. P.D. designed the study and wrote the manuscript. H.J.S. designed study and wrote the manuscript.

**Competing interests.** We declare we have no competing interests. P.D. is currently on sabbatical at Uber AI Lab.

**Funding.** This work was supported by the Wellcome Trust (grant no. 094850/Z/10/Z) and James S. McDonnell Foundation to H.J.S., and the Gatsby Charitable Foundation (P.D.).

**Acknowledgements.** We thank Mate Lengyel for advice on the experimental design.

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
