## [Reviewer comments · Proceedings of the Royal Society B: Biological Sciences]

Review History

RSPB-2019-0476.R0 (Original submission)

Review form: Reviewer 1 (Blake Johnson)

Recommendation

Major revision is needed (please make suggestions in comments)

Scientific importance: Is the manuscript an original and important contribution to its field?

Acceptable

General interest: Is the paper of sufficient general interest?

Acceptable

Quality of the paper: Is the overall quality of the paper suitable?

Marginal

Is the length of the paper justified?

Yes

Should the paper be seen by a specialist statistical reviewer?

Yes

Do you have any concerns about statistical analyses in this paper? If so, please specify them explicitly in your report.

No

It is a condition of publication that authors make their supporting data, code and materials available - either as supplementary material or hosted in an external repository. Please rate, if applicable, the supporting data on the following criteria.

Is it accessible?

N/A

Is it clear?

N/A

Is it adequate?

N/A

Do you have any ethical concerns with this paper?

No

Comments to the Author

This manuscript reports an exploratory re-analysis of MEG and fMRI measured while participants performed a virtual navigation task. The manuscript is generally clear and well-written, the topic is interesting and the re-analysis provides a number of interesting results that may be useful in guiding the design of future studies. I do have a number of suggestions for improvement that can be considered in a revision.

I like the idea of analysing spontaneous behaviours that were not foreseen in the main experimental design but that point to useful and potentially interesting cognitive mechanisms. However the motivation for the re-analysis could use some beefing up. The authors provide very little background on backtracking and why it might be important, apart from a mention of this behaviour in ants. The Ekstrom et al. (2018) citation is not listed in the bibliography.

It is unclear to me why the authors employed a between group design here, with two separate groups studied with different imaging techniques (fMRI and MEG). A within subject design is generally more powerful and would have enhanced the fMRI/MEG comparisons (and as side benefit would have also provided structural MRI's that would have strongly enhanced the MEG analysis). The description of the training sessions only mentions preparation for the fMRI session. Some clarification of this issue is needed.

Some minor issues:

CTF just uses the initials in the company name.

Some confusing descriptions e.g. "We found a significant effect of change type in both fMRI and MEG" when referring to analysis of behavioural data.

A few minor typos.

Review form: Reviewer 2

Recommendation

Major revision is needed (please make suggestions in comments)

Scientific importance: Is the manuscript an original and important contribution to its field?

Good

General interest: Is the paper of sufficient general interest?

Good

Quality of the paper: Is the overall quality of the paper suitable?

Acceptable

Is the length of the paper justified?

Yes

Should the paper be seen by a specialist statistical reviewer?

No

Do you have any concerns about statistical analyses in this paper? If so, please specify them explicitly in your report.

No

It is a condition of publication that authors make their supporting data, code and materials available - either as supplementary material or hosted in an external repository. Please rate, if applicable, the supporting data on the following criteria.

Is it accessible?

Yes

Is it clear?

Yes

Is it adequate?

Yes

Do you have any ethical concerns with this paper?

No

Comments to the Author

The authors present data linking dACC activity to backtracking relative to other navigational event types in a dynamic spatial decision-making task. Moreover, they show evidence that “default mode” regions classically involved in navigation (e.g., the hippocampus) are less active in these circumstances.

They also demonstrate that alpha-band power from MEG is reduced prior to initiation of a backtracking event, consistent with attention to competing actions.

I am a fan of this paradigm development, and I certainly agree with the authors that backtracking is a critically under-studied aspect of navigation behavior.

I have several concerns and points for clarification

1) On p. 9, the authors report predictions about “prefrontal and anterior cingulate regions” and

conducting region-of-interest analyses, but they do not motivate any specific predictions in the Introduction or detail which ROIs were employed in the Methods. It becomes apparent over the breadth of the manuscript that dACC was an area of particular interest – although it was not studied using a targeted ROI. Is it accurate that the sole ROI analysis conducted was a post-hoc network-level DMN ROI? How were the ROIs defined?

I feel the authors could provide more motivation in the Introduction (and relevant methods details) for their focus on the functional anatomy discussed in the Discussion. Why were areas they frequently target in their work (e.g., Hippocampus, Caudate, DLPFC) not ROIs, not to mention the dACC?

2) Is it accurate that there were approximately 20 “optimal” backtrack trials per person (calculated from the percentages in Table 1 off of 120 trials total). What was the minimum “optimal” backtrack trial count in the sample? My apologies if I overlooked this, but did the authors collapse across optimal and incorrect backtracking trials for the fMRI and EEG analyses?

3) Correct backtracks could be a moderate percentage (61%) of a small percentage of overall trials (18.7%) for False Shortcuts Away, or a larger percentage (71%) of a larger percentage of overall trials (35.9%) for Detour +8. This means the overall activity attributed to backtracks is going to be weighted towards some event manipulations more than others; and activity for non-backtracks is going to be weighted towards different event manipulations. It is difficult to estimate those weightings, because the authors go on to split those into turn- and no-turn event types (which I fully agree with). This leads me to wonder to which situations the reported backtrack vs turn+no-backtrack comparisons should be attributed. As a toy example with two event manipulation types (Detour +8 and Shortcut -4), is it possible that activity estimates for backtracks wind up being weighted towards Detour +8 and correct “turn” events towards Shortcut -4?

It would be helpful to know how turn and no-turn trial counts were, on average, distributed across correct route choices for different event manipulation types. I worry that dividing no-backtrack trials into turn vs no-turn (which I do think is critical for the study) could further exacerbate the weighting mismatch for different conditions between backtracks and no-backtracks.

4) As a broader comment, the authors do not report how non-backtrack events were modeled for the reported analyses (presumably also a 0s stick function?). What predictors were in the design matrix? Were different event manipulations (Detour +8, etc) kept as separate predictors, and they were contrasted with backtracks using a flat weighting scheme? Perhaps contrast weighting or covariate schemes could be employed at such an aggregation stage to combat comment 3

5) Although differences between saccade variance were not found between backtracking and non-backtracking, was this true specifically within 1000ms time window immediately prior to the backtrack turn (MEG time window of interest)? Was this true when subdividing the non-backtracking trials into turns and no-turns?

Decision letter (RSPB-2019-0476.R0)

27-Mar-2019

Dear Dr Patai:

I am writing to inform you that your manuscript RSPB-2019-0476 entitled "Back-tracking during navigation shows enhanced anterior cingulate activity and suppression of alpha oscillations and

'default-mode' brain activity" has, in its current form, been rejected for publication in Proceedings B.

This action has been taken on the advice of referees, who have recommended that substantial revisions are necessary. With this in mind we would be happy to consider a resubmission, provided the comments of the referees are fully addressed. However please note that this is not a provisional acceptance.

Sincerely,

Dr. Sarah F. Brosnan
Editor, Proceedings B
mailto: proceedingsb@royalsociety.org

Associate Editor
Board Member: 1
Comments to Author:

Both reviewers are positive about your manuscript but both also raise some concerns. On the basis of their comments, I must recommend rejection. Having said that, I encourage you to resubmit your manuscript and to address the reviewers' concerns.

Reviewer(s)' Comments to Author:

Referee: 1

Comments to the Author(s)

This manuscript reports an exploratory re-analysis of MEG and fMRI measured while participants performed a virtual navigation task. The manuscript is generally clear and well-written, the topic is interesting and the re-analysis provides a number of interesting results that

may be useful in guiding the design of future studies. I do have a number of suggestions for improvement that can be considered in a revision.

I like the idea of analysing spontaneous behaviours that were not foreseen in the main experimental design but that point to useful and potentially interesting cognitive mechanisms. However the motivation for the re-analysis could use some beefing up. The authors provide very little background on backtracking and why it might be important, apart from a mention of this behaviour in ants. The Ekstrom et al. (2018) citation is not listed in the bibliography.

It is unclear to me why the authors employed a between group design here, with two separate groups studied with different imaging techniques (fMRI and MEG). A within subject design is generally more powerful and would have enhanced the fMRI/MEG comparisons (and as side benefit would have also provided structural MRI's that would have strongly enhanced the MEG analysis). The description of the training sessions only mentions preparation for the fMRI session. Some clarification of this issue is needed.

Some minor issues:

CTF just uses the initials in the company name.

Some confusing descriptions e.g. "We found a significant effect of change type in both fMRI and MEG" when referring to analysis of behavioural data.

A few minor typos.

Referee: 2

Comments to the Author(s)

The authors present data linking dACC activity to backtracking relative to other navigational event types in a dynamic spatial decision-making task. Moreover, they show evidence that "default mode" regions classically involved in navigation (e.g., the hippocampus) are less active in these circumstances.

They also demonstrate that alpha-band power from MEG is reduced prior to initiation of a backtracking event, consistent with attention to competing actions.

I am a fan of this paradigm development, and I certainly agree with the authors that backtracking is a critically under-studied aspect of navigation behavior.

I have several concerns and points for clarification

1) On p. 9, the authors report predictions about "prefrontal and anterior cingulate regions" and conducting region-of-interest analyses, but they do not motivate any specific predictions in the Introduction or detail which ROIs were employed in the Methods. It becomes apparent over the breadth of the manuscript that dACC was an area of particular interest – although it was not studied using a targeted ROI. Is it accurate that the sole ROI analysis conducted was a post-hoc network-level DMN ROI? How were the ROIs defined?

I feel the authors could provide more motivation in the Introduction (and relevant methods details) for their focus on the functional anatomy discussed in the Discussion. Why were areas they frequently target in their work (e.g., Hippocampus, Caudate, DLPFC) not ROIs, not to mention the dACC?

2) Is it accurate that there were approximately 20 "optimal" backtrack trials per person (calculated from the percentages in Table 1 off of 120 trials total). What was the minimum "optimal" backtrack trial count in the sample? My apologies if I overlooked this, but did the authors collapse across optimal and incorrect backtracking trials for the fMRI and EEG analyses?

3) Correct backtracks could be a moderate percentage (61%) of a small percentage of overall trials (18.7%) for False Shortcuts Away, or a larger percentage (71%) of a larger percentage of overall trials (35.9%) for Detour +8. This means the overall activity attributed to backtracks is going to be weighted towards some event manipulations more than others; and activity for non-backtracks is going to be weighted towards different event manipulations. It is difficult to estimate those weightings, because the authors go on to split those into turn- and no-turn event types (which I fully agree with). This leads me to wonder to which situations the reported backtrack vs turn+no-backtrack comparisons should be attributed. As a toy example with two event manipulation types (Detour +8 and Shortcut -4), is it possible that activity estimates for backtracks wind up being weighted towards Detour +8 and correct "turn" events towards Shortcut -4?

It would be helpful to know how turn and no-turn trial counts were, on average, distributed across correct route choices for different event manipulation types. I worry that dividing no-backtrack trials into turn vs no-turn (which I do think is critical for the study) could further exacerbate the weighting mismatch for different conditions between backtracks and no-backtracks.

4) As a broader comment, the authors do not report how non-backtrack events were modeled for the reported analyses (presumably also a 0s stick function?). What predictors were in the design matrix? Were different event manipulations (Detour +8, etc) kept as separate predictors, and they were contrasted with backtracks using a flat weighting scheme? Perhaps contrast weighting or covariate schemes could be employed at such an aggregation stage to combat comment 3

5) Although differences between saccade variance were not found between backtracking and non-backtracking, was this true specifically within 1000ms time window immediately prior to the backtrack turn (MEG time window of interest)? Was this true when subdividing the non-backtracking trials into turns and no-turns?

Author's Response to Decision Letter for (RSPB-2019-0476.R0)

See Appendix A.

RSPB-2019-1016.R0

Review form: Reviewer 1 (Blake Johnson)

Recommendation

Accept as is

Scientific importance: Is the manuscript an original and important contribution to its field?

Good

General interest: Is the paper of sufficient general interest?

Good

Quality of the paper: Is the overall quality of the paper suitable?

Good

Is the length of the paper justified?

Yes

Should the paper be seen by a specialist statistical reviewer?

No

Do you have any concerns about statistical analyses in this paper? If so, please specify them explicitly in your report.

No

It is a condition of publication that authors make their supporting data, code and materials available - either as supplementary material or hosted in an external repository. Please rate, if applicable, the supporting data on the following criteria.

Is it accessible?

Yes

Is it clear?

Yes

Is it adequate?

Yes

Do you have any ethical concerns with this paper?

No

Comments to the Author

All concerns have been addressed.

Review form: Reviewer 2

Recommendation

Accept with minor revision (please list in comments)

Scientific importance: Is the manuscript an original and important contribution to its field?

Good

General interest: Is the paper of sufficient general interest?

Good

Quality of the paper: Is the overall quality of the paper suitable?

Good

Is the length of the paper justified?

Yes

Should the paper be seen by a specialist statistical reviewer?

No

Do you have any concerns about statistical analyses in this paper? If so, please specify them explicitly in your report.

No

It is a condition of publication that authors make their supporting data, code and materials available - either as supplementary material or hosted in an external repository. Please rate, if applicable, the supporting data on the following criteria.

Is it accessible?

Yes

Is it clear?

Yes

Is it adequate?

Yes

Do you have any ethical concerns with this paper?

No

Comments to the Author

I thank the authors for their thorough response to my comments. I believe the manuscript is much more clear and my concerns have been largely addressed.

I still have a lingering concern from my previous review that I am not sure has been wholly addressed (previously comment 3).

To restate the comment:

""""3) Correct backtracks could be a moderate percentage (61%) of a small percentage of overall trials (18.7%) for False Shortcuts Away, or a larger percentage (71%) of a larger percentage of overall trials (35.9%) for Detour +8. This means the overall activity attributed to backtracks is going to be weighted towards some event manipulations more than others; and activity for non-backtracks is going to be weighted towards different event manipulations. It is difficult to estimate those weightings, because the authors go on to split those into turn- and no-turn event types (which I fully agree with). This leads me to wonder to which situations the reported backtrack vs turn+no-backtrack comparisons should be attributed. As a toy example with two event manipulation types (Detour +8 and Shortcut -4), is it possible that activity estimates for backtracks wind up being weighted towards Detour +8 and correct "turn" events towards Shortcut -4?

It would be helpful to know how turn and no-turn trial counts were, on average, distributed across correct route choices for different event manipulation types. I worry that dividing no-backtrack trials into turn vs no-turn (which I do think is critical for the study) could further exacerbate the weighting mismatch for different conditions between backtracks and no-backtracks. """"

In response, the authors further bolstered their argument that backtrack activity differs from Turn activity. These are certainly helpful data for the manuscript, but I don't believe this addresses my concern.

To try to clarify the above comment further: is it possible that in a given participant, backtrack events could correspond to more Detour +8 trials than other trial types, e.g. Shortcut -4? And conversely for non-backtrack events to be represented differently?

It seems from Table 1 that this is likely true. In which case, how do we know a backtrack vs no-backtrack activity difference in a person might not actually reflect the fact that activity for that condition was estimated from a regressor built from different proportion of trials from experimenter manipulated trial types (Detour +8, Shortcut -4, etc)?

Decision letter (RSPB-2019-1016.R0)

14-Jun-2019

Dear Dr Patai

I am pleased to inform you that your manuscript RSPB-2019-1016 entitled "Back-tracking during navigation shows enhanced anterior cingulate activity and suppression of alpha oscillations and 'default-mode' brain activity" has been accepted for publication in Proceedings B pending one minor revision. Reviewer 2 has raised an additional question that you will need to address. Because the schedule for publication is very tight, it is a condition of publication that you submit the revised version of your manuscript within 7 days. If you do not think you will be able to meet this date please let us know.

- 1) A text file of the manuscript (doc, txt, rtf or tex), including the references, tables (including captions) and figure captions. Please remove any tracked changes from the text before submission. PDF files are not an accepted format for the "Main Document".
- 2) A separate electronic file of each figure (tiff, EPS or print-quality PDF preferred). The format should be produced directly from original creation package, or original software format. PowerPoint files are not accepted.
- 3) Electronic supplementary material: this should be contained in a separate file and where possible, all ESM should be combined into a single file. All supplementary materials accompanying an accepted article will be treated as in their final form. They will be published alongside the paper on the journal website and posted on the online figshare repository. Files on

figshare will be made available approximately one week before the accompanying article so that the supplementary material can be attributed a unique DOI.

Sincerely,

Dr Sarah Brosnan

Associate Editor

Board Member

Comments to Author:

We have now heard back from our reviewers about your revised manuscript. I am pleased to say that I am willing to recommend acceptance of your manuscript -- but only if you can deal with

the remaining concern of Reviewer 2, which he/she raised in the review of your original manuscript.

Reviewer(s)' Comments to Author:

Referee: 1

Comments to the Author(s).
All concerns have been addressed.

Referee: 2

Comments to the Author(s).
I thank the authors for their thorough response to my comments. I believe the manuscript is much more clear and my concerns have been largely addressed.

I still have a lingering concern from my previous review that I am not sure has been wholly addressed (previously comment 3).

To restate the comment:

""3) Correct backtracks could be a moderate percentage (61%) of a small percentage of overall trials (18.7%) for False Shortcuts Away, or a larger percentage (71%) of a larger percentage of overall trials (35.9%) for Detour +8. This means the overall activity attributed to backtracks is going to be weighted towards some event manipulations more than others; and activity for non-backtracks is going to be weighted towards different event manipulations. It is difficult to estimate those weightings, because the authors go on to split those into turn- and no-turn event types (which I fully agree with). This leads me to wonder to which situations the reported backtrack vs turn+no-backtrack comparisons should be attributed. As a toy example with two event manipulation types (Detour +8 and Shortcut -4), is it possible that activity estimates for backtracks wind up being weighted towards Detour +8 and correct "turn" events towards Shortcut -4?

It would be helpful to know how turn and no-turn trial counts were, on average, distributed across correct route choices for different event manipulation types. I worry that dividing no-backtrack trials into turn vs no-turn (which I do think is critical for the study) could further exacerbate the weighting mismatch for different conditions between backtracks and no-backtracks. ""

In response, the authors further bolstered their argument that backtrack activity differs from Turn activity. These are certainly helpful data for the manuscript, but I don't believe this addresses my concern.

To try to clarify the above comment further: is it possible that in a given participant, backtrack events could correspond to more Detour +8 trials than other trial types, e.g. Shortcut -4? And conversely for non-backtrack events to be represented differently?

It seems from Table 1 that this is likely true. In which case, how do we know a backtrack vs no-backtrack activity difference in a person might not actually reflect the fact that activity for that condition was estimated from a regressor built from different proportion of trials from experimenter manipulated trial types (Detour +8, Shortcut -4, etc)?

Author's Response to Decision Letter for (RSPB-2019-1016.R0)

See Appendix B.

Decision letter (RSPB-2019-1016.R1)

08-Jul-2019

Dear Dr Patai

I am pleased to inform you that your manuscript entitled "Back-tracking during navigation shows enhanced anterior cingulate activity and suppression of alpha oscillations and 'default-mode' brain activity" has been accepted for publication in Proceedings B.

Open Access

You are invited to opt for Open Access, making your freely available to all as soon as it is ready for publication under a CC BY licence. Our article processing charge for Open Access is £1700. Corresponding authors from member institutions (<http://royalsocietypublishing.org/site/librarians/allmembers.xhtml>) receive a 25% discount to these charges. For more information please visit <http://royalsocietypublishing.org/open-access>.

Paper charges

Sincerely,

Dr Sarah Brosnan
Editor, Proceedings B
mailto: proceedingsb@royalsociety.org

Associate Editor:

Comments to Author:

I am delighted to recommend acceptance of your revised manuscript. Congratulations on a fine piece of research.

Appendix A

We thank the Reviewers for the constructive feedback and suggestions – we have addressed all issues raised and have revised our manuscript accordingly. We have highlighted our responses in bold and also indicated where text has been added to the manuscript (additions/changes are highlighted in yellow).

Referee: 1

Rev 1, Point 1

“This manuscript reports an exploratory re-analysis of MEG and fMRI measured while participants performed a virtual navigation task. The manuscript is generally clear and well-written, the topic is interesting and the re-analysis provides a number of interesting results that may be useful in guiding the design of future studies. I do have a number of suggestions for improvement that can be considered in a revision.

I like the idea of analysing spontaneous behaviours that were not foreseen in the main experimental design but that point to useful and potentially interesting cognitive mechanisms. However the motivation for the re-analysis could use some beefing up. The authors provide very little background on backtracking and why it might be important, apart from a mention of this behaviour in ants. The Ekstrom et al. (2018) citation is not listed in the bibliography.”

Thank you for the positive response, and we agree, more motivation for backtracking behaviour would be beneficial. We have now added Ekstrom et al (2018) to the bibliography and have added this text to the Introduction:

“...In these instances, a good strategy is to turn around and back-track to a previous point and begin an efficient route. Such spontaneous and internally-generated behaviour is anecdotally common, and reports of human behaviour that include backtracking often discuss it in the context of lost individuals (Hill, 1998), specifically how backtracking is a part of explorative behaviour and the flexible use of landmarks (Karimpur, Röser, & Hamburger, 2016) (‘looking back’ has been shown to aid subsequent navigation by allowing the navigator to see the environment from different perspectives, i.e. forming an allocentric map), and how the choice to initiate a back-track is related to confidence levels (Hill, 1998). Additionally, back-tracking is a strategy that is not only common but integral to navigation in visually impaired people (Flores & Manduchi, 2018), and even sighted people have been reported to dislike when not given the option to back-track (D’Orazio & Lueg, 2012). It is not clear if back-tracking behaviour is related to navigation ability (Cornell, Heth, & Rowat, 1992; Ruddle, 2001), though older people do show a deficit in retracing their steps (Wiener, Kmecova, & de Condappa, 2012), and navigators often return to a previous decision point and perform a breadth-first search of spaces –i.e. of upcoming the path options (Streeter & Vitello, 1986), especially if they are skilled (Hill, 1998). Finally, this behaviour may be universal in animal navigation, as it has also been reported in ants, which use back-tracking as part of specific search strategy when displaced from their nest (Wystrach, Schwarz, Baniël, & Cheng, 2013), and is implemented in many robotics applications (Tanaka, Zha, & Hasegawa, 1999). Despite the wealth of observational reports on back-tracking, how the brain supports such back-tracking behaviour during goal-directed navigation remains elusive, and it is unclear under what conditions this (usually) corrective behaviour arises.”

Rev 1, point 2

“It is unclear to me why the authors employed a between group design here, with two separate groups studied with different imaging techniques (fMRI and MEG). A within subject design is generally more powerful and would have enhanced the fMRI/MEG comparisons

(and as side benefit would have also provided structural MRI's that would have strongly enhanced the MEG analysis). The description of the training sessions only mentions preparation for the fMRI session. Some clarification of this issue is needed.”

The reviewer is correct that a within subject design would have been very beneficial. However, our experiment involved a learning task followed by a recall task with prediction errors (detours / shortcuts). This would make it likely that participants would have adapted to expect the detours /shortcuts during the second scan. Thus, we opted to conduct the research on 2 groups with the same exposure and experiences. Nonetheless, it would be very useful in future research to explore within-subjects.

We apologise if the Methods were not clear: the training and test were identical in both fMRI and MEG tasks. We have now added reference to the MEG scan along with the mention of fMRI testing throughout (e.g. page 6: “Participants were tested over two days, on day one they were trained on the maze, and on day two they were tested on the maze in the MRI/MEG scanner.”)

Rev 1, Minor issues:

CTF just uses the initials in the company name.

Some confusing descriptions e.g. “We found a significant effect of change type in both fMRI and MEG” when referring to analysis of behavioural data.

A few minor typos.”

Thank you for pointing these out, they have all been rectified.

Referee: 2

“The authors present data linking dACC activity to backtracking relative to other navigational event types in a dynamic spatial decision-making task. Moreover, they show evidence that “default mode” regions classically involved in navigation (e.g., the hippocampus) are less active in these circumstances.

They also demonstrate that alpha-band power from MEG is reduced prior to initiation of a backtracking event, consistent with attention to competing actions.

I am a fan of this paradigm development, and I certainly agree with the authors that backtracking is a critically under-studied aspect of navigation behavior.”

We were glad to see this response and grateful for the useful suggestions for improvement below.

Rev 2, point 1:

“I have several concerns and points for clarification

1) On p. 9, the authors report predictions about “prefrontal and anterior cingulate regions” and conducting region-of-interest analyses, but they do not motivate any specific predictions in the Introduction or detail which ROIs were employed in the Methods. It becomes apparent over the breadth of the manuscript that dACC was an area of particular interest – although it was not studied using a targeted ROI. Is it accurate that the sole ROI analysis conducted was a post-hoc network-level DMN ROI? How were the ROIs defined?”

Thank you for pointing out this lack of clarity. We have now made the ROI section of the Methods more clear (pg. 10). The Reviewer is correct, only one ROI was previously used: the putative DMN, which was defined anatomically including the medial PFC, the precuneus/posterior cingulate cortex, bilateral parahippocampal cortex and angular gyrus. We now include three ROIs: the DMN, hippocampus and a dACC ROI. The motivation for these have also been referenced in the Introduction (see response to point 2 below).

“General Linear Models were constructed at the onset on the back-tracking (and non-backtracking, turn and non-turn) events, with a duration of 0 seconds. We report results surviving family-wise error correction (FWE), as well as results of region-of-interest (ROI) analysis of the putative default-mode network (defined anatomically including the medial PFC, the precuneus/posterior cingulate cortex, bilateral parahippocampal cortex and angular gyrus). We obtained the ROI mask from Kaplan et al. (2017) in order to test for involvement of the dorsal anterior cingulate area (dACC). To measure hippocampal response, we used the mask used in our previous study (Patai et al. 2019).”

Rev 2, point 2:

“I feel the authors could provide more motivation in the Introduction (and relevant methods details) for their focus on the functional anatomy discussed in the Discussion. Why were areas they frequently target in their work (e.g., Hippocampus, Caudate, DLPFC) not ROIs, not to mention the dACC?”

Thank you for these suggestions. We have now examined the data with a hippocampal mask from our recent previous study (Patai et al., 2019). The response in this region does not reach significance. We now provided the rationale for why we targeted particular anatomical regions in our Introduction:

“Under the hypothesis that re-orienting during back-tracking would involve re-engagement in the task we predicted a reduction in default-mode activity (Smith, Mitchell, & Duncan, 2018). Based on prior studies reporting self-correction of errors (Gehring & Knight, 2000; Walton, Devlin, & Rushworth, 2004; for review see Botvinick, 2007) we predicted prefrontal and anterior cingulate regions would show increased activity during back-tracking. Relatedly, one study found that the right posterior hippocampus was engaged when a switch in the path was needed at forced detours (Howard et al. 2014), thus it is possible that backtracking may engage this region due to re-estimation of the future path (see Spiers and Barry 2015).”

And Results (page 15):

Follow-up ROI analysis revealed a significant effect in the dACC (small-volume correction: $Z=5.91$; $p<0.001$), and no significant effects in the hippocampus.

Rev 2, Pont 3:

“(2) Is it accurate that there were approximately 20 “optimal” backtrack trials per person (calculated from the percentages in Table 1 off of 120 trials total). What was the minimum “optimal” backtrack trial count in the sample? My apologies if I overlooked this, but did the authors collapse across optimal and incorrect backtracking trials for the fMRI and EEG analyses?”

There were about 30 backtracking events per participant (pg 8) and of these about 70% were optimal. We included all backtracking events for the fMRI and MEG analyses to achieve sufficient statistical power, but indeed it would be useful in future

studies which encourage more backtracking to differentiate these two outcomes.

Rev 2, Point 4:

“3) Correct backtracks could be a moderate percentage (61%) of a small percentage of overall trials (18.7%) for False Shortcuts Away, or a larger percentage (71%) of a larger percentage of overall trials (35.9%) for Detour +8. This means the overall activity attributed to backtracks is going to be weighted towards some event manipulations more than others; and activity for non-backtracks is going to be weighted towards different event manipulations. It is difficult to estimate those weightings, because the authors go on to split those into turn- and no-turn event types (which I fully agree with). This leads me to wonder to which situations the reported backtrack vs turn+no-backtrack comparisons should be attributed. As a toy example with two event manipulation types (Detour +8 and Shortcut -4), is it possible that activity estimates for backtracks wind up being weighted towards Detour +8 and correct “turn” events towards Shortcut -4?”

It would be helpful to know how turn and no-turn trial counts were, on average, distributed across correct route choices for different event manipulation types. I worry that dividing no-backtrack trials into turn vs no-turn (which I do think is critical for the study) could further exacerbate the weighting mismatch for different conditions between backtracks and no-backtracks.”

Thank you for raising this issue. The turn vs Non-Turn effects we report are based on 1. All such events and 2. Matched to the Backtracking trials. We have updated our Methods (page 9) and Results section (page 16) to clarify:

“For follow-up analyses we also looked at onset of Turn (and Non-Turn) events, which for one analysis were taken as all such events in the experiment (and subsampled to match one another in trial numbers), and another model in which we took these events from the non-backtracking trials and thus matched event numbers with this condition. As a follow-up, we also took Turns from backtracking trials only, (in order to have comparative distribution amongst different trial types). This is helpful in providing evidence that the effects observed for back-tracking were, for example, not simply driven by comparing events sampled in periods after forced-detours with other events sampled in trials with no detours.”

“We found that back-tracking significantly suppressed this putative DMN compared with Turns ($Z=4.62$, $p=0.006$, and $Z=4.37$, $p=0.018$ for comparison within backtracking trials only), and this was also the case when comparing back-tracking to non-back-tracking events ($Z=4.21$, $p=0.038$), but not when comparing Non-Turns to Turns ($p>.1$), highlighting disengagement from this network when participants spontaneously instigated a route change (Figure 3, Table 2). When comparing Turns and Non-Turns, with only matched number of events (to backtracking), we find no significant effects between the conditions, further underscoring the magnitude of the backtracking effects seen despite limited trial numbers.”

“4) As a broader comment, the authors do not report how non-backtrack events were modelled for the reported analyses (presumably also a 0s stick function?). What predictors were in the design matrix? Were different event manipulations (Detour +8, etc) kept as separate predictors, and they were contrasted with backtracks using a flat weighting scheme? Perhaps contrast weighting or covariate schemes could be employed at such an aggregation stage to combat comment 3”

All analyses in this report were based on a GLM containing only the backtracking and non-backtracking events, modelled with a stick function (see page 10 and response to point 1 above).

“5) Although differences between saccade variance were not found between backtracking and non-backtracking, was this true specifically within 1000ms time window immediately prior to the backtrack turn (MEG time window of interest)? Was this true when subdividing the non-backtracking trials into turns and no-turns?”

We thank the reviewer for this suggestion, we re-examined saccadic activity in the pre-backtracking period (-700 to -150 when the alpha difference was reported) and found no significant effect. We now report this on page 16:

“Moreover, the change in alpha activity could not be explained by differences in saccadic activity across the trial types (whole trial or during the significant pre-backtrack time-window, $t(1,23) < -.9$, $p > .1$).”

Appendix B

We thank the Reviewers and the Editor for the positive response to our previous modifications of the manuscript. We have responded here to the final concern of Reviewer 2 and added a note of this additional analysis in the manuscript (highlighted in yellow).

Referee: 2

To restate the comment:

3) Correct backtracks could be a moderate percentage (61%) of a small percentage of overall trials (18.7%) for False Shortcuts Away, or a larger percentage (71%) of a larger percentage of overall trials (35.9%) for Detour +8. This means the overall activity attributed to backtracks is going to be weighted towards some event manipulations more than others; and activity for non-backtracks is going to be weighted towards different event manipulations. It is difficult to estimate those weightings, because the authors go on to split those into turn- and no-turn event types (which I fully agree with). This leads me to wonder to which situations the reported backtrack vs turn+no-backtrack comparisons should be attributed. As a toy example with two event manipulation types (Detour +8 and Shortcut -4), is it possible that activity estimates for backtracks wind up being weighted towards Detour +8 and correct "turn" events towards Shortcut -4?

It would be helpful to know how turn and no-turn trial counts were, on average, distributed across correct route choices for different event manipulation types. I worry that dividing no-backtrack trials into turn vs no-turn (which I do think is critical for the study) could further exacerbate the weighting mismatch for different conditions between backtracks and no-backtracks."""

In response, the authors further bolstered their argument that backtrack activity differs from Turn activity. These are certainly helpful data for the manuscript, but I don't believe this addresses my concern.

To try to clarify the above comment further: is it possible that in a given participant, backtrack events could correspond to more Detour +8 trials than other trial types, e.g. Shortcut -4? And conversely for non-backtrack events to be represented differently?

It seems from Table 1 that this is likely true. In which case, how do we know a backtrack vs no-backtrack activity difference in a person might not actually reflect the fact that activity for that condition was estimated from a regressor built from different proportion of trials from experimenter manipulated trial types (Detour +8, Shortcut -4, etc)?

We thank the Reviewer for clarifying the request, and we fully agree that this is a very important and crucial point. We have now gone back to our data and chosen non-backtracking events from *within* the same trials as in which backtracking occurs – in this way we can fully control for the potential 'state' differences. However, of course we are not able to apply the step matching control. Given the robustness of the control analysis results below, we hope the Reviewer will agree that our effect is valid and interesting to report.

We present our results on the following page:

The frontal effect (dACC) in the backtracking vs non-backtracking survives small-volume correction, as does the DMN when looking at the opposite contrast.

backtracking

non-backtracking

We have added to the Methods section (page 9): “Finally, we also took a new set of non-backtracking events from the same trials as those in which backtracking did occur, excluding any steps near the beginning, end, or near the change point.”

And Results (page 15 and 16):

“When controlling for trial type (detour, shortcut, etc.), i.e. looking at back-tracking compared to non-backtracking events within the same trial, we replicate the significant effect of the dACC ROI (small-volume correction: $Z=4.6$; $p<0.001$).”

“...and this was also the case when comparing back-tracking to non-back-tracking events ($Z=4.21$, $p=0.038$, and using within trial control: $Z=4.77$, $p=0.003$),...”